# Quantification of the Risk of Musculoskeletal Disorders of the Upper Limb Using Fuzzy Logic: A Study of Manual Wheelchair Propulsion

**DOI:** 10.3390/s23218659

**Published:** 2023-10-24

**Authors:** Claire Marchiori, Dany H. Gagnon, Didier Pradon

**Affiliations:** 1Forvia, Faurecia, Automotive Seating, 91150 Brières-les-Scellés, France; claire.marchiori@forvia.com; 2Fondation Garches, 92380 Garches, France; 3Pathokinesiology Laboratory, Centre for Interdisciplinary Research in Rehabilitation of Greater Montreal, Institut de Réadaptation Gingras-Lindsay-de-Montréal, Montreal, QC H3S 2J4, Canada; dany.gagnon.2@umontreal.ca; 4School of Rehabilitation, Université de Montréal, Montreal, QC H3S 2J4, Canada; 5Pôle Parasport CHU Raymond Poincaré, APHP, 92380 Garches, France; 6U1179 Endicap, UVSQ, 78000 Versailles, France

**Keywords:** wheelchair, biomechanics, musculoskeletal disorders

## Abstract

Background: For manual wheelchair users, overuse of the upper limbs can cause upper limb musculoskeletal disorders, which can lead to a loss of autonomy. The main objective of this study was to quantify the risk level of musculoskeletal disorders of different slope propulsions in manual wheelchair users using fuzzy logic. Methods: In total, 17 spinal cord injury participants were recruited. Each participant completed six passages on a motorized treadmill, the inclination of which varied between (0° to 4.8°). A motion capture system associated with instrumented wheels of a wheelchair was used. Using a biomechanical model of the upper limb and the fuzzy logic method, an Articular Discomfort Index (ADI) was developed. Results: We observed an increase in articular discomfort during propulsion on a slope with increasing discomfort at the shoulder, elbow and wrist, due to an increase in kinetics. There was a kinetically significant change in the kinetic global ADI (22 to 25%) and no change in the kinematic. The ADI increased from 14 to 36% during slope propulsion for each joint. Conclusion: The quantification of the level of discomfort helps us to highlight the situations with the most high-risk exposures and to identify the parameters responsible for this discomfort.

## 1. Introduction

Different terms are used for talking about musculoskeletal disorders (MDs) (musculoskeletal symptoms or discomfort), but with the same meaning. The first symptoms are pain, suffering, discomfort and/or numbness in the flat skeletal muscle. Over the last ten years, the number of MDs has increased in all industrial countries. For example, in Europe, 20 to 45% of workers were diagnosed with a neck or upper limb MD during the last twelve months [1]. This is a significant issue in terms of cost, and prevention plans are required in every affected sector [2]. To improve the preventative measures, it is crucial to identify and understand the risk factors. Ergonomic studies have shown that the risk factors are mostly biomechanical, including effort, repetition and extreme joint postures [3,4].

Another, less obvious, group that is affected by MD is users of assistive technologies. Impaired body functions, such as limited walking ability, may lead to restricted activity and participation [4] as mobility is a prerequisite for participation in societal life [5]. Assistive technologies have been developed to increase participation [6]. Manual wheelchairs (MWCs) are the principal mode of mobility for many people with lower limb disabilities. According to the World Health Organization (WHO), the number of people with disabilities is estimated at 1% of the world’s population. In France, 62 persons per 10,000 inhabitants—i.e., 360,000 persons—use a wheelchair, 59 out of 10,000 of whom use a MWC [7]. Achieving a high degree of independence with a MWC often depends on the user’s ability to negotiate a range of environments and to overcome obstacles both indoors and outdoors, such as slopes. Slopes are often the means of accessing buildings, facilitating social integration and participation. In France, the 2005 accessibility act stated that all public buildings must be accessible to persons in wheelchairs within ten years. It specified that slopes should be less than 5% (2.8°) with a plateau every 10 m; however, slopes of up to 8 and 10% (4.6 and 5.7°) could be tolerated if the plateau is placed, respectively, every 2 and 0.5 m. However, in reality, limited space means that slopes are sometimes steeper. To date, only a few studies have investigated the efforts required by the upper limbs (UL) during uphill propulsion. It has been reported that a substantial number of manual wheelchair users (MWCUs) are unable to ascend steep slopes: 85–88% of MWCUs can ascend an 8% slope [8,9]. This number decreases as the slope increases. Moreover, uphill propulsion places additional demands on the user’s UL, which further increases the risk of secondary musculoskeletal impairments, especially at the wrists and shoulders [10,11]. The propulsion of a wheelchair up a 5% slope requires 46% of the user’s maximal isometric propulsive force [12], and for a 10% slope, requires 61% of the user’s maximal isometric propulsive force. It is thus not surprising that the velocity decreases by about 1.5 and 2.7 times for slopes of 3° and 6°, respectively, in comparison with level ground [13]. Despite the reduction in speed, the peak total force decreases by about 1.5 and 2.2 times when ascending a 3° and 6° slope. Yang et al. (2012) showed that the total and tangential forces applied at the handrim were 2.09- to 2.38-times higher when pushing up a 3° slope in comparison to a level surface. These studies demonstrate that uphill propulsion places significant stress on UL joints [14]. 

Increased stress on UL joints may lead to the development of musculoskeletal disorders. The quantification of the risk of MD, including the factors involved, could help to improve the design of MWCs in order to reduce MDs in MWCUs. However, discomfort and MD risk during a motor task is difficult to quantify, and most ergonomic scales are subjective. Various methods have been proposed, including the evaluation of the kinematics and kinetics of propulsion [15,16,17]. Putz-Anderson et al. (1988) stressed the importance of the risk factors’ synergy. It thus seems essential to develop a comprehensive and multi-factorial index for the analysis of the risk of MD during up-slope propulsion in MWCUs [18].

The purpose of this study was therefore to propose a method for evaluating the risk of MD in MWC users. We designed an “Articular Discomfort Index” (ADI) that incorporates ergonomic knowledge and clinical expertise. The aim was to enable the analysis of the specific impact that the degree of slope could have on the displacement of the MWC and on the joints in order to propose improvements in the design of MWCs. We hypothesized that the risk of MD would increase with the steepness of the slope.

## 2. Materials and Methods

### 2.1. Participants

Seventeen individuals with spinal cord injury (SCI) between C4 and T12 were recruited (Table 1). The inclusion criteria were complete or incomplete SCI (American Spinal Injury Association Impairment Scale (AIS) = A, B or C) at least three months before inclusion, use of a MWC for more than 4 hours per day and independent community mobility in the MWC, including ascending an access ramp. Participants were excluded if they presented other neurological conditions or U/L musculoskeletal impairments/pain or cardiorespiratory/vascular conditions or any other impairments or disabilities that might have interfered with the performance or the safety of the experimental tasks. Approval was obtained from the Research Ethics Committee of the Centre of Interdisciplinary Research in Rehabilitation of Greater Montreal (2015 CRIR #715-0312). Participants were informed about the nature of the study before signing informed consent.

### 2.2. Protocol

#### 2.2.1. Clinical Assessment

Each participant underwent a clinical assessment with a physical therapist. Personal characteristics (age, dominant arm, time since injury, wheelchair experience, etc.) and anthropometric parameters (height, weight) were recorded. The American spinal cord injury association impairment scale (AIS) was used to characterize the severity of the sensory and motor impairments. The Wheelchair User’s Shoulder Pain Index (WUSPI) [19] was used to confirm that no incapacitating musculoskeletal impairment was affecting their ULs.

#### 2.2.2. Laboratory Assessment

Self-selected natural MWC propulsion speed was assessed in a 20 m long corridor. Three trials were carried out with a two-minute rest between each. The mean of the three tests was calculated and used to select the speed of the treadmill for the next test.

Participants propelled their own MWC on a dual-belt instrumented treadmill (width = 0.84 m; length = 1.84 m) specially adapted for MWC propulsion (Bertec, Columbus, OH, USA). A familiarization period at different speeds and slopes from those studied was allowed prior to recording. Each participant then completed two trials of up to one minute at their self-selected natural speed, measured on the ground for each of the four treadmill slopes: 0°, 2.7°, 3.6° and 4.8°. The four slopes greater than 0° corresponded to slopes that increase from one unit of height to 20, 16, 12 and 8 units of length, respectively. The order of the trials was randomized, except for the task at 0°, which was always carried out first for technical reasons (laboratory calibration). A two-minute rest was given between each trial. 

UL and wheelchair kinematics were recorded using a motion analysis system (Optotrak 100 Hz model 3020; NDI TechnologyInc., Waterloo, ON, Canada). This system tracked the three-dimensional (3D) coordinates of 23 marker LEDs fixed on the skin over specific anatomical landmarks and tracked four additional LEDs attached to the wheelchair frame in order to record the 3D motion of the upper extremities, trunk and wheelchair (Figure 1). Additionally, specific anatomical body landmarks and wheelchair reference points were also digitized to further define the principal axes of segments and joint centers. Marker trajectories were filtered using a fourth-order zero-lag Butterworth filter with a cut-off frequency of 6 Hz. Segmental coordinate systems (head, trunk, arms, forearms and hand) were defined according to the ISB recommendations [19]. The relative motion of each segment of the upper limbs was computed using the joint coordinate system (JCS) method [20].

Participants used their own MWCs, which were equipped with two instrumented wheels that recorded the forces and moments applied to the handrims at 240 Hz (Smartwheel: Three River Holdings, Mesa, AZ, USA).

The Smartwheel system is composed of a handrim with three beams of instrumented strain gauge bridges connected to an acquisition board, allowing the generation of six channels (Ch1, Ch2, Ch3, Ch4, Ch5, Ch6). The three beams are spaced 120° apart and are fixed on the wheel hub and on the handrim. The handrim is not fixed to the wheel rim. This instrumentation makes it possible to quantify the mechanical forces exerted by the hand on the handrim. In addition to these three beams, an optical angular encoder makes it possible to quantify the rotation of the wheel, and therefore, to know the position of each beam. The beams are spaced 120° apart and a mathematical model was applied to calculate the three force components (Fx, Fy, Fz) and the three torque components (Mx, My, Mz) [21]. This mathematical model takes into account: the radius of the handrim (R), the measurement of the beams (m) and its calibration constant (k), as well as the angle (Ɵ) of Channel 1, which is considered as a reference.
Fx = kCh1 * mCh1 * cos (Ɵ) + kCh2 * mCh2 * co s(120° + Ɵ) + kCh3 * mCh3 * cos (240° + Ɵ)
Fx = kCh1 * mCh1 * sin (Ɵ) + kCh2 * mCh2 * sin (120° + Ɵ) + kCh3 * mCh3 * sin (240° + Ɵ)
Fz = kCh4 * mCh4 + kCh5 * mCh5 + kCh6 * mCh6
Mx = kCh4 * mCh4 * R * cos (Ɵ) + kCh5 * mCh5 * R * cos (120° + Ɵ) + kCh6 * mCh6 * R * cos (240° + Ɵ)
My = kCh4 * mCh4 * R * sin (Ɵ) + kCh5 * mCh5 * R * sin (120° + Ɵ) + kCh6 * mCh6 * R * sin (240° + Ɵ)
Mz = kCh4 * mCh4 * R + kCh5 * mCh5 * R + kCh6 * mCh6 * R

Synchronization of the SmartWheels with the Optotrak motion analysis system was conducted by synchronizing the peak vertical force produced simultaneously on both handrims with an instrumented hammer, off line. Forces and moments were first filtered using a fourth-order-zero-lag Butterworth filter with a cut-off frequency of 20 Hz and then down-sampled to 30 Hz to align with the kinetic data. Only the data of the non-dominant U/Ls were analyzed because it was expected that the non-dominant U/L would be the most likely to limit performance during a functional task requiring symmetrical bilateral efforts, such as propelling a MWC on a linear trajectory [22].

An inverse dynamics method was used to estimate net joint moments of the non-dominant wrist, elbow and shoulder joints with respect to the laboratory coordinate system [23]. A MATLAB algorithm using the forces applied at the handrim, UL kinematics and anthropometric parameters based on the mass and height of each subject was used to compute these net joint moments [24]. Net joint moments and kinematics at the wrist, elbow and shoulder joints were expressed within their respective JCS.

MWC propulsion was divided into two distinct phases: the push phase and the recovery phase, and the mean duration of each was calculated. The push angle was calculated as the difference between the initial and the final angle of the non-dominant hand on the handrim of the MWC. 

### 2.3. The Articular Discomfort Index

A 3-step fuzzy logic method was used to create and evaluate the Articular Discomfort Index (ADI). 

Step one: the fuzzification converts numeric values of the system inputs (kinematics (angles) and kinetics (moments)) to the different degrees of the membership fuzzy sets of partition risk: “low”, “medium” and “high”. The partition discomfort values were based on previous studies in the literature [15,16,17,25,26,27,28,29,30,31,32,33,34].

Step two: the inference module consists of two blocks; namely, the inference engine and inference rules. The rules and their weightings were defined based on the knowledge of several experts (physiotherapist, clinicians, occupational therapist, etc.).

Step three: the defuzzification inferred a specific value of MD risk from the result of the aggregation rules.

The system inputs were kinetics (moments) and kinematics (angles) and the output was the global ADI. Kinetics were calculated in two steps: the risk of MD for each degree of freedom; then, the risk of MD of the joint. For the first step, we chose five fuzzy intervals and triangular and trapezoidal type membership functions. The range of the universe of discourse varies with the amplitude of the anatomical extent of freedom. The predicates “low”, “medium” and “high” were selected. The value of the step one output was used as the input for step two. Three fuzzy intervals and triangular types of membership functions were chosen. For the moment inputs, the generated joint moment amplitude was created. Then, the amplitudes of each joint were normalized by the mass of the system (MWCU mass + MWC mass in kg). The normalized net time varies between 0 and 100% of the theoretical maximum moment of the joint. We chose three fuzzy intervals and the same membership function type. For the generated moment amplitude, the range of the universe of discourse varies between 0 and 100%, representative of the moment percentage relative to the maximum moment. Concerning the output, three fuzzy intervals membership functions of the triangular type were chosen. The same predicates for the inputs and output were selected: “low”, “medium” and “high”. A literature review was carried out to determine the risk factors for MD (kinematics and kinetics) and the values associated in the workplace and related to wheelchair use. The rules and their weightings are defined based on the expertise of several experts. The DELPHI method and the results of the literature search were used to create the rules [15,16,17,25,26,27,28,29,30,31,32,33,34,35].

For step two, the inference method chosen was the Mamdani method. Therefore, the AND is performed by calculating the minimum, while the operator OR is performed by calculating the maximum. 

The last step was carried out using the center of gravity method based on the calculation of the abscissa corresponding to the surface centroid of the fuzzy subset of the solution, determined by the aggregating action of fuzzy rules.

The ADI consists of two elements: global and focused. The global ADI provides an assessment of the risk of MD for each slope. The focused ADI is then calculated for each degree of freedom of each joint. The focused ADI thus provides information that can be used to reduce the risk of MD.

### 2.4. Statistical Analysis

The means and standard deviations of the demographic and clinical data of the participants (Table 1) and all the main outcome measures (Table 2) were calculated. To verify the effects of increasing the treadmill slope (slopes of 0°, 2.7°, 3.6°and 4.8°) on the variables of interest, a Freidman ANOVA was used with the significance level set at *p* < 0.05. When the differences were significant, Wilcoxon tests were carried out (post hoc tests) with an adjusted significance level set at *p* < 0.05. Correlation coefficient and mean squared errors (RMSE) were then performed between the different slopes in order to quantify and characterize the changes. A threshold of 11% was set for the RMSE (established according to the significant achievements of Wilcoxon test) and 0.75 for the correlation coefficient.

## 3. Results

All of the participants were able to propel their MWC on level ground and on the treadmill at 0°, as well as up slopes of 2.7°, 3.6° and 4.8°. The mean self-selected comfortable propulsion speed was 1.17 ± 0.18 m/s [min = 0.91 m/s; max = 1.65 m/s].

### 3.1. Spatiotemporal Propulsion Parameters

#### 3.1.1. Propulsion Phases

Table 2 summarizes the mean duration of the push and recovery phases and the total duration of a propulsion cycle in seconds for the different slopes under analysis. The mean duration of the push phase was shorter for the slopes than at 0°, but there was no difference between the durations for the 2.7° and 4.8° slopes. The mean duration of the recovery phase decreased significantly with the increasing slope, except between the 2.7° and 4.8° slopes. The mean propulsion cycle duration decreased with the increasing slope. The differences between the 0° condition and the other slopes were significant.

#### 3.1.2. Push Angle

The initial push angle decreased significantly with the increase in the slope, except between 2.7° and 3.6 ° (no significant difference). The final push angle was greater for the slopes than the 0° condition (*p* ≤ 0.001), but was not different between slopes. The total push angle did not differ between slopes (*p* = 0.135 to 0.241), except that it was greater for the 2.7° slope than the 4.8° slope (*p* = 0.004).

#### 3.1.3. Propulsion Pattern

The method of propulsion in the 0° condition differed between participants (Table 1). It also generally changed with the increasing slope, with some participants using “single loops” or “double loops”. The “arc” style was the most used (23 to 47% of participants).

#### 3.1.4. Trunk Kinematics

The trunk flexion increased between propulsions in the 0° condition and each slope (from 16 to 27° during the push phase and from 11 to 19° during the recovery phase) (Figure 2).

### 3.2. MD Analysis

Table 3 and Table 4 summarize the risk of MD calculated for the non-dominant UL for the different slopes. The global and focused Articular Discomfort Indexes (ADI) are shown in Table 5 and Figure 2.

#### 3.2.1. Global Articular Discomfort Index

The ADI showed a medium risk of MD for the 0° condition (64.5 ± 25.8%), as well as for the 2.7° and 3.6° slopes (44.8 ± 22.1%). The risk was high for the 4.8° slope (49.7 ± 20.6%). The global ADI during the push phase (18–20%) was greater for the slopes than the 0° condition (*p* < 0.001). There was a significant increase in the kinetic global ADI (22 to 25%, *p* = 0.06 to 0.3) and no change in the kinematic global ADI (*p* = 0.001 *p* = 0.003). Only the correlation coefficient of the kinematic global ADI was not significant (r < 0.75).

No significant change appeared between the slope levels in the global ADI (*p* = 0.3 to 0.8). There were no significant differences in the kinematic and kinetic global ADIs between slopes, except for the kinetic ADI between the 2.7° and the 4.8° slopes (*p* = 0.01).

#### 3.2.2. Focused Articular Discomfort Index

The focused ADI increased from 14 to 36% for the slopes compared with the 0° condition for each joint (*p* = 0.0001 to 0.04). The correlation coefficient was only significant for the shoulder (r > 0.9). The focused ADI of the elbow and wrist increased by 12% from the 2.7° to the 4.8° slopes (*p* = 0.04 and *p* = 0.08). The coefficient was significant for the wrist and shoulder between slope levels (r > 0.94).

Shoulder:

The focused kinematic and kinetic ADIs increased significantly for the three degrees of freedom of the shoulder from the 0° condition to the different slopes (14 to 36% *p* = 0.0001 to 0.04). There was a significant difference in the RMSE between the 0° condition and the 4.8° slope for flexion–extension and internal–external rotation. There was no change in the shoulder kinematics for the focused ADI (*p* = 0.06 to 0.3). There was a significant correlation between all three DOFs and the focused ADI of the shoulder (r > 0.9).

There was no significant difference between the kinematic focused ADI for the three slopes (*p* > 0.5). There was a significant difference between the kinetic focused ADI for the 2.7° and 4.8° slopes (*p* = 0.03). The correlation coefficient for each DOF was significant. The correlation for the kinematic focused ADI was only significant between the 2.7° and 3.6° slopes.

Elbow:

There were significant differences between the 0° condition and the slopes for the kinematic and kinetic focused ADIs (*p* < 0.002 for kinematics and *p* = 0.0001 to 0.04 for kinetics). The correlation coefficient was not significant for the kinematic focused ADI, but was significant for the kinetic focused ADI. There was a significant difference between the focused ADI of the 3.6° and the 4.8° slopes (*p* = 0.04). There was a significant difference in the kinetic focused ADI between the 2.7° and 4.8° slopes (*p* = 0.04). The correlation coefficient of the kinetic focused ADI was significant (r > 0.9).

Wrist:

There was no significant difference in the kinematic focused ADI between the 0° condition and the slopes (*p* > 0.5); however, there was a significant difference for the radio-ulnar deviation (*p* < 0.02). The kinetic focused ADI was significant for each DOF (*p* = 0.0001 to 0.01). The RMSE of the kinematics and the radio-ulnar deviation was not significant, indicating no change in the risk of MD. The RMSE of the kinetic focused ADI and the two DOFs was significant. The kinetic focused ADI increased from 27 to 35%. The correlation coefficients were significant for the two DOFs for the kinematic and kinetic focused ADIs (r > 0.92).

There were no significant differences in the kinematic and kinetic focused ADIs between slopes, except for the difference in the kinetic focused ADI of the two DOFs, which was significant between 2.7° and 4.8°. The RMSE of the kinematic focused ADI and the correlation coefficient of the kinematic focused ADI of the wrist were not significant. The coefficient of two DOFs was significant.

Recovery Phase

The correlation coefficients of the global ADI and the kinematic and kinetic global ADIs were significant (r > 0.77). The correlations were significant at the elbow and wrist for all conditions (r > 0.82), while for the shoulder, they were only significant between different slope levels (r < 0.75 between ground level and the different slopes and r > 0.97 between the slopes). There were no significant differences in the RMSE for the global ADI or for the kinematic and kinetic global ADI in any condition. For the analysis of each joint, only the RMSE of the shoulder was significant between ground level and the two highest degrees of slope (11% discomfort for 3.6° and +15% to 4.8°).

The correlation coefficient of the kinematic focused ADI was significant for all of the joints and DOFs (r > 0.83). The RMSE was not significant, showing no change in the risk of MD. The correlation coefficients of the three joints were not significant (r < 0.6); however, they were significant for each DOF, except for shoulder abduction–adduction and internal–external rotation. The RMSE was only significant for the shoulder, with an increase in the risk of MD from 14 to 30% according to the DOF.

## 4. Discussion

The aim of this study was to characterize and quantify the risk of MD during propulsion up different angles of slopes. We hypothesized that the Articular Discomfort Index (ADI) would increase with the slope and that the increase would be due to an increase in both kinematic and kinetic discomfort. Furthermore, we believe that the joint discomfort in the shoulder and wrist are mainly responsible for the ADI increase.

The global ADI increased from 18 to 20% during propulsion up the slopes compared with the 0° condition. This is consistent with previous studies of the kinematics and kinetics of MWC propulsion up slopes. Several studies have shown that upper limb joint pain can occur during a variety of activities of daily living (ADL), but that it is much more intense during certain activities, such as propulsion up slopes and transfers [19,36,37]. A study in paraplegic subjects carrying out ADL involving extreme amplitudes of the shoulder or significant amounts of force showed that up-slope propulsion was one of the most painful activities. This was particularly the case for persons with tetraplegia [19].

As hypothesized, the focused ADI was greater for the slopes than the 0° condition for each joint. The increase was greatest at the wrist (29 to 36%), followed by the elbow (16 to 26%), then the shoulder (14 to 19%). There was no difference between the slopes. This increase was mainly due to an increase of 22–29% in the kinetic focused ADI during the push phase. This suggests that greater efforts are required to propulse the wheelchair from the smallest slope (2.7°), but that this effort does not increase further with the increasing slope. The kinematic focused ADI tended to increase with the slope, although this was not significant. It was greatest at the wrist, followed by the shoulder, then the elbow. The high frequency of repetition of such high-risk movements may explain the high prevalence levels of MD in MWCUs, particularly at the wrist and shoulder.

The results of the kinetic focused ADI are consistent with the data from the literature. van Drongelen et al., in 2005, showed that propulsion up a slope multiplied the peak of net moment at the shoulder and elbow by two compared with propulsion on a flat surface (7.2 ± 2.4 vs. 14.6 ± 3.8 Nm for the shoulder and 3.0 ± 2.3 vs. 5.7 ± 2.1 Nm for the elbow) [38]. The efforts made during up-slope propulsion are high from a very shallow slope and changes little with the increasing steepness. Our results showed that the kinematic ADI increased in each joint during up-slope propulsion. Although the MWCUs altered their posture in the W/C on the slopes, this did not cause a significant increase in the risk of discomfort at the wrist and shoulder. The changes in the kinematic risk of MD at the elbow were mainly due to the kinematic focused ADI for flexion–extension. These results likely relate to the particular technique or propulsion style of the user, the inclination of the trunk and the push angle. The results showed that MWCUs who used “single loop” and “double loop” techniques on flat ground mainly changed to an “arc” style for the slope, with a trunk bent forward posture. This is consistent with previous studies. Qi et al. (2013) also showed that users’ propulsion style changes with the slope and tends towards an “arc” style [39]. The increase in trunk flexion maintains the center of mass within the lift base, defined by the four wheels being on the ground. The wheelchair used in the present study was a dynamic MWC. This relates to the ease of caster pop to overcome obstacles. This creates a degree of instability in the MWC, which users compensate for by adapting their posture. It was thus not surprising that the MWCUs leaned forward with the increasing slope to prevent backwards tilting of the MWC. The decrease in the duration of the recovery phase during up-slope propulsion minimizes the deceleration of the MWC. The deceleration increased as the slope increased. As during synchronous propulsion, both hands exert a force on the handrim at the same time, there is no mechanical action on the handrim during the recovery phase. Therefore, the MWC is only subjected to forces that slow its progression. Our results showed that when the slope increased, the user reduced the duration of the recovery phase in order to limit backwards motion of the wheelchair on the treadmill.

The trunk flexion increased during up-slope propulsion and was associated with a change in the initial and final push angles on the handrim. Up-slope, these angles were slightly offset towards the front of the handrim (relative to the vertex of the wheel) and the push angle was reduced. These parameters did not change with the increase in the slope. The forward shift of the push angle is in agreement with the results of Chow et al. (2009). These authors suggested that the increase in trunk flexion during up-slope propulsion may cause the hand to move towards the front of the handrim [40]. Reducing the push angle may be a strategy to minimize UL fatigue and MD risk during up-slope propulsion [41].

The propulsion style was also affected by the increase in the slope. The results of our study showed that the participants tended to change their style of propulsion to an “arc” style with the increasing slope. Qi et al. (2013) showed that this style, which maintains the hands closer to the handrim, is associated with a shorter recovery phase [39]. Our results showed a decrease of approximately 70% in the duration of the recovery phase, depending on the slope (1.01 s at 0° and 0.58 s on the 4.8° slope), while the duration of the push phase remained similar in the flat condition and across the slopes. These results are like those found in the literature [42]. This likely reflects an adaptation strategy to limit the deceleration of the MWC when there is a lack of propulsive force. 

Although there was a decrease in the duration of the recovery phase up-slope, the overall risk of MD remained unchanged, with the ADI values remaining below 25%. However, it is important, and rated as High, over the entire cycle. The increase in the ratio of the push phase to the recovery phase increases the ‘at risk’ time of the joints, while reducing the recovery time. This reduction in recovery time can increase the risk of MD. Similarly, the increased frequency of propulsion up-slope has also been related to the risk of MD in the UL joints [10].

The shoulder kinetic ADI also increased slightly (14%). This increase was mainly caused by the increase in the focused ADI for shoulder flexion–extension. This increase is likely related to the segmental movement speed of the arm. Indeed, the reduction in the duration of the recovery phase caused an increase in the upper limb movement speed combined with a likely increase in muscle activity [41]. Qi et al. previously showed that the activity of the principal muscles that were active during the recovery phase increased significantly with the propulsion speed [39].

The use of the ADI to quantify MD risk during propulsion on different angles of slopes demonstrated differences between the 0° condition and up-slope propulsion. Some differences were also revealed between shallower (2.7°) and steeper slopes (4.8°). These results demonstrate that an increased risk of MD occurred in two steps: firstly, when a slope was introduced; secondly, when the gradient became much steeper.

Figure 2 shows that the MD risk is greater as the inclination increases and is not constant throughout the propulsion cycle. This evolution of the MD risk is in line with various studies quantifying, in particular, the EMG activity and forces of the muscles involved in incline displacement [43,44]. Indeed, Dany et al. indicate that these muscles are activated at different times during the propulsion cycle, but that the EMG activity also increases as the slope increases. Holoway et al. report that, for certain shoulder muscles, the muscular force, calculated from musculoskeletal models, increases as the slope increases. In addition to this work focusing on muscular contribution, other studies have quantified oxygen consumption [45]. These studies show that as the slope increases, so does the oxygen consumption.

All of these studies quantifying the various muscular and energy parameters involved in wheelchair use on slopes highlight the need to find technical or technological solutions for users. We believe that a better understanding of the risk of MD, as we propose to quantify it, will enable clinical teams to raise wheelchair users’ awareness of the need to maintain sufficient physical condition and muscular strength to maintain independent displacement. We also believe that when this risk of MD evolves unfavorably, the use of electric propulsion assistance could increase this autonomy of movement by reducing the force produced by the user, as well as his or her oxygen consumption [46].

Study limitations:

This study has some limitations. First, there was a large degree of inter-subject variability in some of the kinematic parameters because of the different propulsion styles used. However, although the variability is high, the increase in risk remains similar from one subject to another. The impact of propulsion style could not be investigated because of the small number of subjects. It may be appropriate to perform this study again on a larger scale. We plan to carry out a study using an ergometer powered roller with increasing resistance in 62 persons with paraplegia in order to explore some or all four of the styles of propulsion.

The participants used their own MWCs for the study; thus, the W/C configuration parameters varied between participants. Finally, some of the participants reported cardiorespiratory fatigue during the tasks, which may have affected their performance. Further studies should therefore include an evaluation of the physiological demands required during up-slope propulsion on a treadmill.

## 5. Conclusions

The results of this study showed that up-slope propulsion increased the risk of MD (change of style, decreased recovery phase, increased efforts, etc.). The results were consistent and reinforced those of existing studies. More specifically, the risk of MD increased in all three joints and was mainly related to increased kinetics. 

This study showed that the ADI is useful for quantifying the risk of MD in upper limb joints and to determine the kinematic, kinetic and combined aspects of the risk. This can be used to determine the most high-risk situations and to identify causative parameters. Such measures are significant because they can lead to recommendations, such as those regarding slope steepness for entering a building.

## Figures and Tables

**Figure 1 sensors-23-08659-f001:**
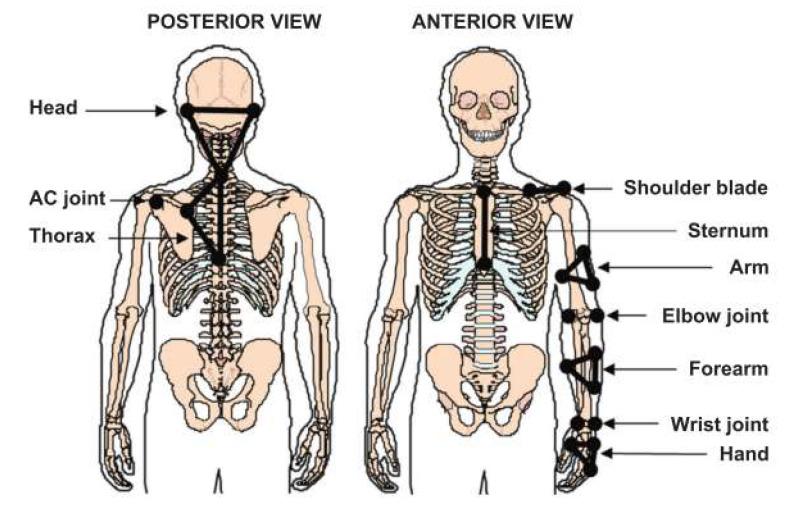
Schematic representation of the location of the skin-fixed infrared LEDs used as markers on the subject’s segments.

**Figure 2 sensors-23-08659-f002:**
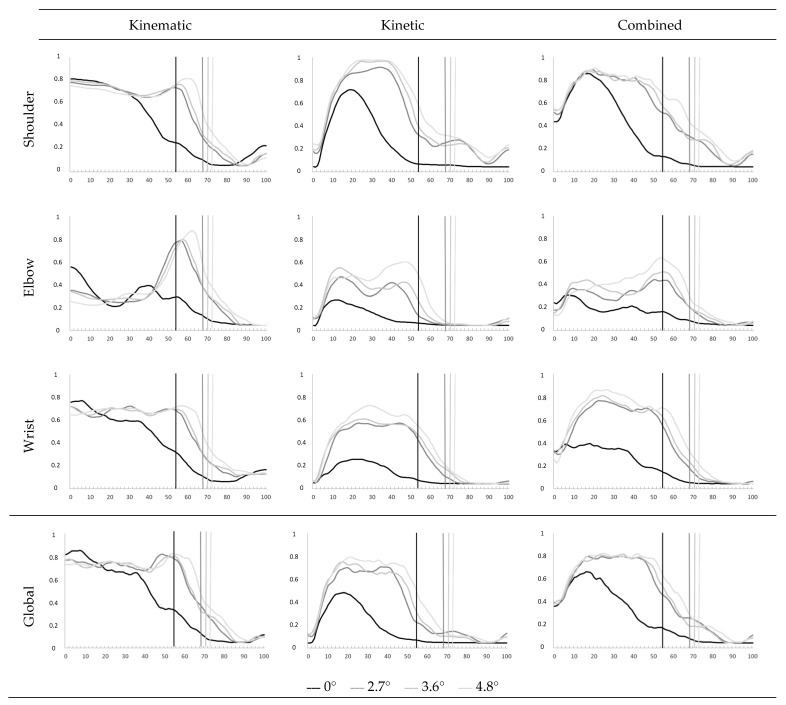
Graphical representation of the global and focus Articular Discomfort Index (ADI) propulsion on different slope levels during the propulsion cycle. The demarcation between the push and return phase for each slope level is represented by a vertical line of the same color as the ADI.

**Table 1 sensors-23-08659-t001:** Characteristics of participants. * Gender: M = Men. W = Women; WUSPI: Wheelchair User’s Shoulder Pain Index, sc = semi-circular, arc = arcing, slop = single looping over propulsion, dlop = double looping over propulsion.

N°	Gender *	Age (Years)	Time Since Injury (Years)	Height (m)	Weight (kg)	Dominance	Lesion Level	WUSPI * Score	Pattern			
									sc	arc	slop	dlop
1	M	22.3	0.7	1.83	57.5	Right	T8	0.29	X			
2	M	58.3	3.4	1.88	98.9	Right	T10	0.26			X	
3	M	42.8	9.1	1.84	86	Right	T7	0.46		X		
4	M	34.4	10.3	1.80	77.1	Right	T6	0.30			X	
5	M	51.2	7.9	1.73	77.5	Right	T12	0.45		X		
6	M	54.2	3.4	1.80	106.3	Left	T3	0.00		X		
7	W	26.6	3.3	1.65	45.2	Right	T11	0.08			X	
8	M	45.0	3.2	1.73	76.9	Right	T10	0.14	X			
9	M	31.9	6.5	1.93	71.8	Right	T12	0.1	X			
10	M	31.4	7.5	1.75	66.7	Right	T8	0.0				X
11	M	53.0	17.1	1.75	93.0	Right	T10	0.1				X
12	M	30.8	4.0	1.91	93.5	Right	T10	0.0		X		
13	M	27.0	9.2	1.9	67.7	Right	T12	0.0			X	
14	M	40.0	4.8	1.8	75.3	Right	C6	0.0	X			
15	M	42.7	20.8	1.7	65.1	Right	C4	0.0		X		
16	M	47.8	3.1	1.7	81.5	Right	T7	0.3			X	
17	M	37.8	1.5	1.8	97.4	Left	T9	0.1			X	
**mean**		**39.8**	**6.8**	**179**	**78.7**			**0.2**				
**SD**		**10.8**	**5.4**	**0.1**	**16**			**0.2**				

**Table 2 sensors-23-08659-t002:** Measures of Times phases of cycle propulsion for each slop and push angle (start. end. total) (mean ± standard deviation) with a percentage comparison to 0°. Definition of propulsion pattern. *: *p* < 0.05 between 0° and each slope.

		0°	2.8°	3.6°	4.2°
**Spatio-temporal parameters**					
Push Phase	(s)	0.55 ± 0.1 ***	0.44 ± 0.1	0.43 ± 0.1	0.42 ± 0.1
	(%)	54.4	67.5	69.3	72.5
Recovery Phase	(s)	0.46 ± 0.2 ***	0.21 ± 0.1	0.19 ± 0.1	0.16 ± 0.1
	(%)	45.6	32.5	30.7	27.5
Cycle Time	(s)	1.01 ± 0.2 ***	0.65 ± 0.1	0.63 ± 0.1	0.58 ± 0.1
**Push angle**					
start	(°)	−33.00 ± 9.75 *	−19.84 ± 12.02	−17.60 ± 9.70	−11.08 ± 10.20
end	(°)	45.97 ± 9.04 *	52.04 ± 9.20	53.46 ± 10.36	57.92 ± 11.82
Total	(°)	78.97 ± 13.27 *	68.94 ± 22.92	69.90 ± 14.77	66.91 ± 15.36
**Propulsion pattern**					
Semi circular		4	4	4	4
arc		4	5	8	8
Simple loop		2	5	4	4
Double loop		7	2	1	1

**Table 3 sensors-23-08659-t003:** Percentage (%) of risk level of musculoskeletal disorders for kinematic for each slope during push and recovery phase of the propulsion cycle.

		Risk Level	0.0		2.7		3.6		4.8	
Shoulder Flexion Extension	Push	Low	13.6	4.1	0.7	2.2	1.7	2.7	0.5	2.0
		Medium	74.1	17.4	54.8	4.5	51.3	1.9	54.3	6.3
		High	12.2	15.8	44.5	2.8	47.0	1.9	45.2	4.6
	Recovery	Low	99.2	1.1	21.0	6.6	20.7	6.5	4.7	7.5
		Medium	0.8	1.1	79.0	6.6	79.3	6.5	95.3	7.5
		High	0	0	0	0	0	0	0	0
Shoulder Abduction Adduction	Push	Low	96.7	6.8	43.0	5.2	40.5	5.7	37.0	3.8
		Medium	3.3	6.8	55.7	6.8	59.1	5.9	59.7	7.5
		High	0	0	1.3	4.9	0.5	1.8	3.3	7.9
	Recovery	Low	99.5	1.0	66.8	8.5	73.8	8.6	69.9	7.7
		Medium	0.5	1.0	33.2	8.5	26.2	8.6	29.5	6.5
		High	0	0	0	0	0	0	0.5	2.1
Shoulder Internal External Rotation	Push	Low	11.6	2.8	6.8	1.2	5.1	1.7	0.6	2.1
		Medium	55.3	8.8	57.6	7.4	76.4	5.1	83.2	6.4
		High	33.0	8.0	35.6	7.4	18.5	3.6	16.2	5.1
	Recovery	Low	93.7	5.1	55.6	9.3	52.5	9.0	17.8	9.5
		Medium	6.3	5.1	44.4	9.3	47.5	9.0	82.2	9.5
		High	0	0	0	0	0	0	0	0
Elbow Flexion Extension	Push	Low	55.3	8.2	24.1	4.0	22.7	11.2	17.5	2.6
		Medium	44.7	8.2	75.9	4.0	77.3	11.2	82.5	2.6
		High	0	0	0	0	0	0	0	0
	Recovery	Low	100	0	100	0	99.3	2.5	98.2	4.9
		Medium	0	0	0	0	0.7	2.5	1.8	4.9
		High	0	0	0	0	0	0	0	0
Elbow Pronation Supination	Push	Low	17.4	7.9	3.9	4.5	6.0	10.2	6.2	12.4
		Medium	82.6	7.9	96.1	4.5	94.0	10.2	93.8	12.4
		High	0	0	0	0	0	0	0	0
	Recovery	Low	99.9	0.5	80.0	5.7	82.1	9.2	79.6	9.6
		Medium	0.1	0.5	20.0	5.7	17.9	9.2	20.4	9.6
		High	0	0	0	0	0	0	0	0
Wrist Flexion Extension	Push	Low	64.5	26.1	10.2	1.0	7.7	0.9	6.9	0.8
		Medium	35.5	26.1	87.5	8.5	89.4	10.7	55.7	12.5
		High	0	0	2.3	8.9	2.9	10.5	37.5	12.7
	Recovery	Low	100	0	98.3	1.9	97.7	3.0	97.3	1.8
		Medium	0	0	1.7	1.9	2.3	3.0	2.7	1.8
		High	0	0	0	0	0	0	0	0
Wrist Abduction Adduction	Push	Low	67.9	16.1	26.7	7.5	16.5	5.3	23.8	17.2
		Medium	32.1	16.1	73.3	7.5	83.5	5.3	76.2	17.2
		High	0	0	0	0	0	0	0	0
	Recovery	Low	100	0	100	0	99.7	0.9	99.8	0.7
		Medium	0	0	0	0	0.3	0.9	0.2	0.7
		High	0	0	0	0	0	0	0	0

**Table 4 sensors-23-08659-t004:** Percentage (%) of risk level of musculoskeletal disorders for kinetic for each slope during push and recovery phase of the propulsion cycle.

		Risk Level	0.0		2.7		3.6		4.8	
Shoulder Flexion Extension	Push	Low	13.6	4.1	0.7	2.2	1.7	2.7	0.5	2.0
		Medium	74.1	17.4	54.8	4.5	51.3	1.9	54.3	6.3
		High	12.2	15.8	44.5	2.8	47.0	1.9	45.2	4.6
	Recovery	Low	99.2	1.1	21.0	6.6	20.7	6.5	4.7	7.5
		Medium	0.8	1.1	79.0	6.6	79.3	6.5	95.3	7.5
		High	0	0	0	0	0	0	0	0
Shoulder Abduction Adduction	Push	Low	96.7	6.8	43.0	5.2	40.5	5.7	37.0	3.8
		Medium	3.3	6.8	55.7	6.8	59.1	5.9	59.7	7.5
		High	0	0	1.3	4.9	0.5	1.8	3.3	7.9
	Recovery	Low	99.5	1.0	66.8	8.5	73.8	8.6	69.9	7.7
		Medium	0.5	1.0	33.2	8.5	26.2	8.6	29.5	6.5
		High	0	0	0	0	0	0	0.5	2.1
Shoulder Internal External Rotation	Push	Low	11.6	2.8	6.8	1.2	5.1	1.7	0.6	2.1
		Medium	55.3	8.8	57.6	7.4	76.4	5.1	83.2	6.4
		High	33.0	8.0	35.6	7.4	18.5	3.6	16.2	5.1
	Recovery	Low	93.7	5.1	55.6	9.3	52.5	9.0	17.8	9.5
		Medium	6.3	5.1	44.4	9.3	47.5	9.0	82.2	9.5
		High	0	0	0	0	0	0	0	0
Elbow Flexion Extension	Push	Low	55.3	8.2	24.1	4.0	22.7	11.2	17.5	2.6
		Medium	44.7	8.2	75.9	4.0	77.3	11.2	82.5	2.6
		High	0	0	0	0	0	0	0	0
	Recovery	Low	100	0	100	0	99.3	2.5	98.2	4.9
		Medium	0	0	0	0	0.7	2.5	1.8	4.9
		High	0	0	0	0	0	0	0	0
Elbow Pronation Supination	Push	Low	17.4	7.9	3.9	4.5	6.0	10.2	6.2	12.4
		Medium	82.6	7.9	96.1	4.5	94.0	10.2	93.8	12.4
		High	0	0	0	0	0	0	0	0
	Recovery	Low	99.9	0.5	80.0	5.7	82.1	9.2	79.6	9.6
		Medium	0.1	0.5	20.0	5.7	17.9	9.2	20.4	9.6
		High	0	0	0	0	0	0	0	0
Wrist Flexion Extension	Push	Low	64.5	26.1	10.2	1.0	7.7	0.9	6.9	0.8
		Medium	35.5	26.1	87.5	8.5	89.4	10.7	55.7	12.5
		High	0	0	2.3	8.9	2.9	10.5	37.5	12.7
	Recovery	Low	100	0	98.3	1.9	97.7	3.0	97.3	1.8
		Medium	0	0	1.7	1.9	2.3	3.0	2.7	1.8
		High	0	0	0	0	0	0	0	0
Wrist Abduction Adduction	Push	Low	67.9	16.1	26.7	7.5	16.5	5.3	23.8	17.2
		Medium	32.1	16.1	73.3	7.5	83.5	5.3	76.2	17.2
		High	0	0	0	0	0	0	0	0
	Recovery	Low	100	0	100	0	99.7	0.9	99.8	0.7
		Medium	0	0	0	0	0.3	0.9	0.2	0.7
		High	0	0	0	0	0	0	0	0

**Table 5 sensors-23-08659-t005:** Percentage (%) of risk level of musculoskeletal disorders for kinematics. Kinetics and handrim kinetics for each slope during push and recovery phase of the propulsion cycle.

		Risk Level	0.0		2.7		3.6		4.8	
Kinematic	Push	Low	0	0	0	0	0	0	0	0
		Medium	0	0	0	0	0	0	0	0
		High	100	0	100	0	100	0	100	0
	Recovery	Low	0	0	0	0	0	0	0	0
		Medium	0	0	2.2	6.9	1.1	4.4	0.9	3.5
		High	100	0	97.8	6.9	98.9	4.4	99.1	3.5
Kinetic	Push	Low	0	0	0	0	0	0	0	0
		Medium	48.2	2.2	20.3	3.8	17.9	2.1	11.9	1.7
		High	51.8	2.2	79.7	3.8	82.1	2.1	88.1	1.7
	Recovery	Low	0	0	0.6	2.5	2.0	4.0	0	0
		Medium	0	0	99.0	3.7	98.0	4.0	97.8	5.7
		High	100	0	0.3	1.2	0	0	2.2	5.7
Global	Push	Low	0	0	0	0	0	0	0	0
		Medium	0	0	0	0	0	0	0	0
		High	100	0	100	0	100	0	100	0
	Recovery	Low	0	0	0	0	0	0	0	0
		Medium	0	0	0.6	2.5	0	0	0	0
		High	100	0	99.4	2.5	100	0	100	0

## Data Availability

Not applicable.

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
