# Peer review of "Quantification of the Risk of Musculoskeletal Disorders of the Upper Limb Using Fuzzy Logic: A Study of Manual Wheelchair Propulsion"

_sensors, 2023, doi:10.3390/s23218659_

Round 1

Reviewer 1 Report

Dear authors, all my comments are flagged in the attached manuscript.

Author Response

Rewiever 1

Thank you for this in-depth analysis. We have taken the liberty of « expanding » on some of the answers to your questions (especially the last one) to demonstrate our scientific honesty.

- Line 17 : « … tilt (0° - 4.8°)… » it’s not cleat. I didn't realize it referred to a treadmill until I read the full text.

We reread the summary and nowhere did we indicate that the tests are on a treadmill. Of course we modified the summary so that readers don't discover this important point in the middle of the article.

- Line 21 : « … three joints … »  which joints?

We specified that these were the shoulder, elbow and wrist joints.

- Line 100 : Table 1 SC Arc SLOP DLO. It is necessary to define these acronyms in the caption.

We added the definition of acronyms

- Line 367 : L. 324 "... there was a significant difference for radio-ulnar deviation (p<.02)...". Did you mean to refer to wrist or radio-ulnar deviation?

We refer to the Global ADI, i.e. the two degrees of freedom of the wrist as well as the kinematics and kinetics. To make it easier for readers to understand, we've added a figure (2). This figure provides a better visualization of the evolution of the risk of MD during the propulsion cycle. We believe that a graphical representation of the Global and Focus ADI is more comprehensible than an additional table. And it will also enable us to introduce your next comment.

- Line 436-437 : I consider it important to include in the discussion the relationship of muscle mechanics. The way the discussion is presented, not that it's bad, but it gives the impression that the article addresses a force generation system, as if it were an engine. However, the muscle has its specificities, such as the force-length relationship (Gordon 1966). The greater the angular variation, the greater the variation in muscle length, drastically altering the perception of effort and truly the ability to produce force.

We also have the force-velocity relationship (Hill 1938). In this case, with the increase in the treadmill's slope, the contraction speed also increases, resulting in a decrease in muscular force production capacity. Muscular power should also be considered, which is the product of force and velocity. Well, if you consider some of the information I mentioned here, it will help readers understand the importance of your study and the relationship between your results and muscle physiology.

It's very hard to respond with confidence to your comment, but it's an opportunity to deepen the discussion. To explain this precaution, allow me to explain myself. I've been interested in the link between joint kinematics and muscular biomechanics. Like many of us, I was impressed by the mechanical model proposed by Hill (1938), but also by Scott Delp's thesis manuscript on musculoskeletal modeling. Or the work of Veeger HE and Van der Helm FC on wheelchair propulsion and the upper limb. This curiosity has led us all to adopt tools such as SIMM or, more recently, Opensim, or LifeModeler, BoB, ... to carry out these analyses and simulations. But all these models are based on the scaling of a generic model. Even if these models are clearly described, and we assume their mechanical assumptions, capturing the scapula's movement from external markers generates imprecision in the calculation of its displacement. Quantifying muscle forces from geometric models is also subject to significant bias.

In general, the use of musculoskeletal modeling of the upper limb should be cautious (Morrow M 10.1016/j.jbiomech.2014.09.013 ). I should point out that this remark is in the context of wheelchair propulsion due to the significant movements of the scapula. I'm sorry to have taken the time to write this part. This is not a sign of scientific disrespect. On the contrary, I wish to indicate that it is a scientific wish, but I remain cautious. Perhaps wrongly so.

Nevertheless, I would like to take advantage of your comment to add a section at the end of the discussion on strength and power, indicating in particular the energy cost. Using figure 2 as a starting point, I'd like to explain that the API increases as difficulty increases, and that this is in line with the literature, particularly on VO2 and muscular strength. This allows me to introduce the notion of the muscular strength required to overcome these resistances to progress. This modification will also enable me to respond to reviewer 2's last remark concerning practical applications.

Reviewer 2 Report

Thank you for the opportunity to review this interesting manuscript. The study is about the quantification of the risk level of musculoskeletal disorders of different slopes propulsion in manual wheelchair by fuzzy logic. The topic is extremely interesting and it has been well presented and discussed. The methodology is strong and well structured, all element to replicate the study are present. The limitations are presented. I have only few minor comments:

Line 43: the statistics are about French people. I think, if possible, if the statistics are about the world characteristics

Table 1: please, specify what are SC, Src, SLOP, DLO. I think a part of the imagine is cut, control this.

Please, add the practical application of the study.

Author Response

Reviewer 2

Thank you for the opportunity to review this interesting manuscript. The study is about the quantification of the risk level of musculoskeletal disorders of different slopes propulsion in manual wheelchair by fuzzy logic. The topic is extremely interesting and it has been well presented and discussed. The methodology is strong and well structured, all element to replicate the study are present. The limitations are presented. I have only few minor comments:

Thank you for your analysis, especially the last comment. We wanted to respond to it in agreement with reviewer 1, as we felt it was complementary.

- Line 43: the statistics are about French people. I think, if possible, if the statistics are about the world characteristics

We have included statistics on the world's disabled population. https://www.who.int/publications/i/item/9789241547482.

- Line 100: Table 1: please, specify what are SC, Src, SLOP, DLO. I think a part of the image is cut, control this.

Also at the request of the reviewer 1, we added the definition of acronyms.

We changed the font so that the acronym DLOP is on a single line

- Line 436-437 : Please, add the practical application of the study.

Your point is very well made, and it's true that it's essential for the clinical reader or sports coach to have ideas for applications. I suggest you add this part at the end of the discussion to include it in the last remark of reviewer 1.

Round 2

Reviewer 1 Report

The authors have addressed my comments.  I agree with their answer, and the manuscript is approved.
